# Urban Sensory Gardens with Aromatic Herbs in the Light of Climate Change: Therapeutic Potential and Memory-Dependent Smell Impact on Human Wellbeing

Izabela Krzeptowska-Moszkowicz [ID], Łukasz Moszkowicz [ID] and Karolina Porada *[ID]

Faculty of Architecture, Cracow University of Technology, 31-155 Krakow, Poland; ikrzepto@pk.edu.pl (I.K.-M.); lmoszkowicz@pk.edu.pl (Ł.M.)
* Correspondence: kporada@pk.edu.pl

**Abstract:** The aim of this study was to analyze urban sensory gardens containing aromatic herbs in terms of the plants used in them. The analysis considered the impact of climate change, particularly of higher temperatures, which may affect the character of contemporary urban gardens. The study was planned primarily in the context of the gardens' therapeutic significance to their users. An important part of the work was to analyze how particular aromatic plants are perceived and received by the inhabitants, using the example of one of Poland's largest cities, Kraków, to assess whether they can have an impact on the inhabitants' positive memories and thus improve their well-being. Initially, the plant composition of gardens located in Poland that feature aromatic herbs was analyzed. This was followed by a survey and an analysis of therapeutic gardens using the Trojanowska method as modified by Krzeptowska-Moszkowicz et al. The plant composition analysis of sensory gardens featuring herbs demonstrated that vulnerable plants in the Central European climate are being introduced to urban sensory gardens. In terms of major aromatic plants, it was found that almost every respondent reported the existence of scents that had some form of essential significance associated with personal memories. Considering the important sensory impact of water elements in therapeutic gardens, as well as problems related to the acquisition of drinking water or water used in agriculture or horticulture, the paper also addresses this topic. It was found that the city dwellers who filled in the questionnaire strongly preferred the introduction of more ecological solutions in the gardens related to water use—to collect and use rainwater, e.g., for watering, instead of piped water.

**Keywords:** sensory gardens; climate change; urban green spaces; aromatic herbs; plant smell memory; stress; human wellbeing in urban areas; restorative environments





## 1. Introduction

### 1.1. Sensory Gardens with Herbs as Therapeutic Places in the Urban Tissue

Publicly accessible sensory gardens are a part of urban public greenery [1]. Similarly other well-designed green areas in cities, they can act primarily as restorative gardens if they guide the user to achieving an internal bodily balance (homeostasis) [2], which results in an improvement in one's emotional and physical fitness, even if by merely reducing stress. A sensory garden is a special type of garden, listed among therapeutic gardens, and as such has significant use in therapy [3]. Such gardens should be designed so that humans are able to experience them from up close [4]. Although the primary function of a sensory garden is to affect its users' senses, it is defined in various ways in the literature, with listings of commonly used definitions provided by Husein [5], Krzeptowska-Moszkowicz et al. [6], Wajchman-Świtalska et al. [7]. To us, the sense of a publicly accessible urban sensory garden is expressed most precisely in the definition of the British Sensory Trust, which states that a garden is: "a self-contained area that concentrates a wide range of sensory experiences [ . . . ], such an area, if designed well, provides a valuable resource for a wide range of uses, from education to recreation" [8]. In an urban setting, sensory

gardens can be established for various types of users and are thus intended for different forms of activity [6].

Sensory gardens should be designed with special care so that they can fulfil their assigned therapeutic tasks [3,4,9,10]. Plants are a crucial component of these gardens, with aromatic herbs being particularly of value [3]. Herbs are defined as "any plant with leaves, seeds or flowers used for flavoring, food, medicine, fragrance production, etc." [11]. In ages past, herbs also held spiritual significance—they had a religious, ritual, or symbolic value [12]. This highlights their important and wide ranging effects on people. Many herbs that appeared in Central and Eastern Europe by the Middle Ages and arrived through convents and monastic gardens, were originally prevalent in warmer areas of Europe and Asia [12–14]. They made their way from monastic gardens to lay ones, and to the collective conscious as part of medicinal preparations, seasoning for meals, or in various types of ceremonies, which contributed to their widespread recognition [12]. Accounting for the origins of many such plants, it was not always possible to cultivate them in the soil in the climate of Poland. However, the currently observed climate change could visibly alter the appearance and extend the significance of gardens that feature herbs.

At present, herbs are desirable element of contemporary gardens and play a crucial role in selected therapeutic programs [3]. This is due to the influence of the scent of aromatic herbs on the sense of smell that does not limit itself to the flowers of such plants, as it also includes their vegetative elements, which extends their application scope. A study by Zajadacz et al. [15] noted that scents play a greater role in the spatial orientation of blind people in sensory gardens than they do outside of them. Arslan et al. [16] also discussed the general matters of the application of aromatic herbs in therapeutic gardens. Aromas, including herbal ones, are also crucial for another reason, as the key significance of scents in stimulating memory-dependent bodily functions was observed, allowing us to "transfer the past into the present" [17,18] and to build certain new relations based on previous experiences. This is used in practice in, among other places, sensory gardens near special education facilities—in children's therapy, where a familiar plant scent, especially one associated with home and family, contributes to better performance in spatial orientation tasks and building self-confidence and autonomy [4].

*1.2. Climate Change, Greenery, and Human Health*

Global climate change results in local phenomena that affect specific regions and countries [19]. Rising temperatures affect people who live in cities in an especially negative way, with high summer temperatures having a profoundly negative impact. This problem is further compounded by the so-called urban heat island effect that occurs in urbanized areas [20]. Heat waves, describing long periods of elevated temperature, are another compounding effect of climate change, and were found to have a particularly detrimental effect on human health and wellbeing, especially in seniors [21,22]. Such phenomena have been observed in Kraków, one of Poland's largest cities, among others [23].

The presence of urban greenery can alleviate the negative consequences of high temperatures, for instance by plant transpiration or through the visibly lower heat accumulation of plant-occupied surfaces in comparison to paved surfaces or open areas [24,25]. A green environment can also positively affect the health and wellbeing of urban residents, even via its passive observation [26]. Numerous studies found that humans desire contact with nature and display a preference for a green setting instead of a typically urbanized one when choosing a place to regenerate after mental exhaustion. Being present in a greenery-rich environment was found to facilitate psychological renewal [27,28]. As opposed to stress and the rapid pace of urban life, a garden offers a slow rhythm and an absence of disruptiveness via its changing plants, which is noted by its users, and brings calmness and peace [29]. Studies performed in Great Britain found that, in a big-city setting, visiting urban green areas led to different types of wellbeing benefits than strolling outside the city. It was found that it reduces anxiety [30]. The positive impact of green spaces on various health-related human problems was investigated in numerous studies, which were

collected and briefly discussed by Chiabai et al. [31]. Papers that describe its impact on the mental wellbeing of adults were summarized by Houlden et al. [32]. Urban greenery positively affects the residents of urban areas and can remediate the negative consequences of climate change, yet it is also impacted by such consequences. Positive phenomena in this respect include an extension of the vegetation period both in Poland and other European countries [33], which contributes to gardens appearing attractive for longer periods, and in the case of therapeutic gardens, extends the season during which they can affect their users. In the light of climate change, a simulated temperature increase of 1 °C relative to the final three decades of the twentieth century was performed and showed a significant decrease in the temperate cool region's reach, and an increase in the area of the temperate warm region, along with the appearance of a warm region [19], which can contribute to changes in the composition of species introduced into green areas in individual territories, and can lead to the spread of thermophilic flora, especially in cities [34,35]. The estimated temperature increase is expected to lead to an increased water deficit [21], which is a climate-change consequence that negatively affects plants and gardens.

*1.3. Goal of the Study*

The purpose of the survey study among the residents of a large city was to investigate the way respondents perceived the smells of various herbs and to reach deeper to determine whether specific herbs had positive memory-dependent significance to respondents. The practical applications of the answers to these questions may aid in better understanding the restorative and therapeutic impact of sensory gardens featuring herbs on city residents. Therefore, the study also featured an analysis of the therapeutic potential of existing Polish gardens with sensory features which primarily targeted the sense of smell. The outcome of this analysis was to indicate how their therapeutic scope can be extended. Problems associated with climate change were also investigated. Our work also takes into account the problems associated with climate change, particularly that of rising temperatures in cities, which may affect existing urban gardens.

## 2. Materials and Methods

*2.1. Research Sites*

The study investigated six publicly accessible urban gardens located in Poland (Table 1). The gardens were selected due to aromatic herbs being their major component, in addition to being intentionally designed as fragrant gardens. The selected gardens were relatively well-known domestically, as they had either already appeared in studies on therapeutic gardens, were featured in garden trails, or were located in areas frequently visited by people from various areas of Poland and abroad. Some of the gardens directly referenced the Middle Ages or the Renaissance, periods when the sense of smell was held in high regard. All of the analyzed gardens were contemporary, although some of them were located in areas previously occupied by historical gardens.

Publicly accessible gardens are defined as gardens that are accessible to all visitors and can be visited either during specific hours or are always open. The following urban gardens with sensory features and located in Poland were selected for the study:

1. Frombork: Herbal garden near a historical medieval hospital building belonging to the Holy Spirit Hospital—currently a museum;
2. Kamień Śląski: The S. Kneipp herbal garden—near a Kneipp Institute therapeutic facility [10];
3. Kraków: The herbal garden of the Czapski Pavilion—a museum;
4. Sandomierz: The Garden of Marcin of Urzędów—a garden referencing a medicinal Renaissance garden [36];
5. Kraków: Zapachowo fragrance garden and sensory path at the S. Lem Science Garden [7,37];
6. Solec Zdrój: Educational Path—Aromatherapeutic Avenue—a fragrant garden built by the Solec Zdrój Municipality.

**Table 1.** Overview of the gardens under study.

| | GARDENS FEATURING HERBS | | | | FRAGRANT GARDENS | |
|---|---|---|---|---|---|---|
| | **1** | **2** | **3** | **4** | **5** | **6** |
| Purpose of establishment | Educational, collecting plants used in medicine | Garden used in Kneipp therapy, intended for aromatherapy strolls | Container garden intended to affect the senses and support pollinating insects living in the city | Educational, used to present selected medicinal plants that used to grow in M. of Urzędów's Renaissance medicinal garden | Intended to stimulate the sense of smell, located in a sensory education park that references the thoughts and deeds of M. Kukelhaus | Intended to stimulate the sense of smell, located in a sensory education park that references the thoughts and deeds of M. Kukelhaus |
| Primary target users | Museum visitors | Kneipp Institute patients undergoing therapy | Museum visitors and city residents | Museum visitors | Primarily children and youth | City residents, patients of the nearby spa treatment center |
| Garden composition | Freeform layout, herbs are grown in various places around the garden in marked beds, the path has a freeform outline | Geometric layout, the path runs between square-shaped beds with individual herb species, a pergola covered with creepers runs perpendicular | Geometric layout, composed of a square-shaped lawn and a path along one of its sides, with plant containers alongside it | Geometric composition, modeled after a Renaissance garden, a circular bed occupies a central position | Freeform composition, the path meanders between clusters of aromatic herbs and bushes | Generally geometric composition, large beds divided by slanted paths into quadrangles of varying size |

Poland is in Central Europe, in a temperate climate zone. Poland is noted to have a transitional climate influenced by Western Europe's marine climate and Eastern Europe's continental climate. Additionally, due to the influx of a diverse range of air masses, the Polish climate is characterized by a degree of variance. Poland's territory can be divided into subregions that differ in temperature, the course of the seasons and amount of rainfall that occurs [19,38]. Temperatures, especially temperature extremes, and particularly those in winter, have a significant impact on the plant types cultivated in gardens. Therefore, so-called frost-resistance zones are delineated, with three such zones in Poland, which are taken into consideration during species-composition formulation for gardens. See Figure 1.

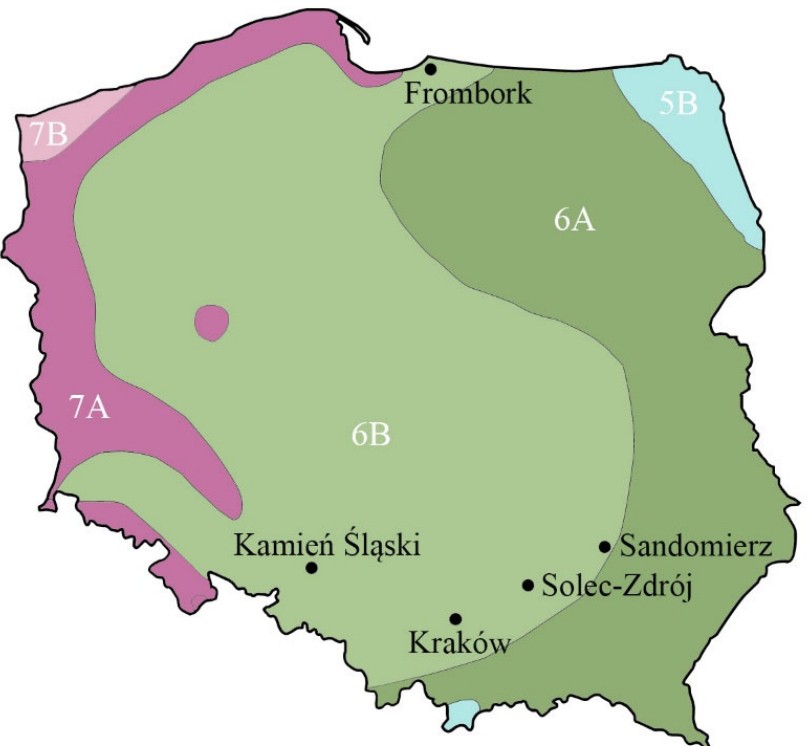

**Figure 1.** This illustration shows a map of Poland with the location of the herb gardens analyzed in this work. Colors and numbers with letters: 5B, 6A, 6B, 7A, 7B indicate frost zones of plants in Poland.

*2.2. Methods*

At the start of the study, the selected plant material present in the gardens was analyzed, with a focus on species that provide olfactory stimulation. This was achieved using the available species lists and with surveys performed in all the gardens. A list of aromatic herbs used in the gardens was compiled. In addition, the species that were listed in the literature as struggling to survive winters in the Polish climate or unable to survive them while planted in the soil were enumerated. The frequency of using temperature-sensitive plants in the sensory gardens under study was investigated, as well as the ways the most vulnerable plants are able to survive winter in urban conditions in the current Polish climate.

A survey study was used to investigate how the individual herb fragrances present in such gardens were perceived by the residents of a large city—Kraków. The survey sample consisted of 73 people. The sample group was small and therefore cannot be considered representative, but in our opinion it is sufficient for the preliminary study described. In future research, the target group of respondents will be expanded.

Seeing as sensory gardens are often dedicated to specific groups of city-dwelling users [6] with distinctive needs, the respondents were divided into the following three age groups: group A—young people between 18 and 29 years of age, typically students

or engaging in first-time employment; group B—people between 30 and 59 years of age, either employed or active in their respective families, group C—seniors aged 60 and above. The respondents were asked six questions, with three concerning aromatic herbs and the remaining three concerning water in herbal gardens. Those aromatic herbs that appeared the most often in Polish gardens with sensory features associated with the sense of smell as well as species sensitive to low temperatures were selected for this study. Concerning water in herbal gardens, the questions in the survey asked whether respondents found it to be significant in herbal gardens, its significance in those gardens and whether the respondents found where said water was sourced from to be significant.

In the first stage, an analysis of the gardens' therapeutic potential was performed using the Trojanowska method [39,40], as adapted to gardens with sensory features by Krzeptowska-Moszkowicz et al. [6]. The method is based on an assessment of the gardens' attributes, whereby an attribute is understood as "a feature of space or the presence of features" [39]. In this method, as the number of attributes increases, so does an area's therapeutic potential. It is a method that has, upon suitable adaptation, been successfully used to analyze therapeutic areas of various sizes, such as sensory gardens [4], parks [39,40] and large areas such as a seacoast fragment with natural and cultural value [41].

The attributes were grouped into design stages as follows: (1) functional program, (2) functional–spatial structure, (3) the design of internal spaces and architectural form, (4) placemaking, (5) accounting for sustainability requirements [39]. The last group of attributes, namely sustainability requirements, is essential for two reasons. First, it brings to mind the problems that appear in green areas, including in response to climate change, and the proper management of natural resources by humans. The second crucial aspect concerns the associated properties of a sensory garden that determine its perception by users.

## 3. Results

### 3.1. Aromatic Herbs in the Gardens with Sensory Features

Table 2 shows the species used in the gardens under study and how often they appeared. The listing includes plants from warmer areas of the world, which are vulnerable during winter, and thus are interesting from a climate-change standpoint. Furthermore, two species of aromatic herbs that appeared less often in these areas while being well-known in Poland were of note (Levisticum officinale W.D.J. Koch and Melissa officinalis L.). The nine species identified were later featured in the survey intended for later stages of the study.

True lavender appeared in all the gardens, in addition to *Salvia officinalis* L. or other species of salvia. Both *Nepeta × Faasenii Bergmans ex Stearn* and *Origanum vulgare* L. were not as prevalent, but still common, as they were present in five cases. Mint, *Mentha x piperita* L., was also used relatively frequently, but mostly peppermint, in addition to *Hyssopus officinalis* L., which was noted in four of the six analyzed gardens. *Melissa officinalis* L., lemon balm, as well as a number of herbs used as dried seasoning in cooking, were rarer, present in only a half of the gardens, which is why they were not included in the survey. Our study showed that other aromatic herbs were only present in gardens in singular cases, and as such cannot be said to be popular as plants used in publicly accessible urban herbal and fragrant gardens.

The group of aromatic herbs that were previously considered not to winter in the soil or only partially winter this way includes as many as four species found in the gardens under study. Interestingly, plants from the following three genera: *Lavandula*, *Salvia* and *Hyssopus*, are very popular in these gardens and were found in either all or nearly all the gardens. Even rosemary, *Rosmarinus officinalis*, which is highly sensitive to low temperatures and was seen as a plant that does not winter in Poland, was present in two gardens. Therefore, it can be stated that sensitive plants were boldly introduced into contemporary urban gardens.

**Table 2.** Presence of aromatic herbs in the gardens under study. The plants typically used in these gardens, as well as more vulnerable plants originating from warmer areas of the world. V is a stamp which should be read as "yes", in this case a confirmation that the plant is in the assigned garden.

| PLANT | RESILIENCE/WINTERING IN POLAND | GARDENS FEATURING HERBS | | | | | | NUMBER OF GARDENS |
|---|---|---|---|---|---|---|---|---|
| | | 1 | 2 | 3 | 4 | 5 | 6 | |
| PERENNIALS THAT WINTER IN POLAND | | | | | | | | |
| *Nepeta × Faasenii Bergmans ex Stearn* | perennial | v | v | v | - | v | v | 5/6 |
| *Origanum vulgare* L. | perennial | v | v | v | v | v | - | 5/6 |
| *Mentha × piperita* L. | perennial | v | v | v | - | v | - | 4/6 |
| Other species and varieties: *Mentha* sp. | | - | v | v | - | v | - | |
| *Melissa officinalis* L. | perennial | v | v | - | - | v | - | 3/6 |
| *Levisticum officinale* W. D. J. Koch | perennial | v | - | v | - | - | - | 2/6 |
| VULNERABLE SPECIES FROM WARMER REGIONS | | | | | | | | |
| *Lavandula angustifolia* Mill. | prostrate shrub/partially winters | v | v | v | v | v | - | 5/6 |
| *Salvia officinalis* L. | subshrub/partially winters | v | v | v | - | v | - | 6/6 |
| Other species: *Salvia* sp. | | - | - | - | v | - | v | |
| *Hyssopus officinalis* L. | subshrub/partially winters | v | v | - | v | v | - | 4/6 |
| *Rosmarinus officinalis* L. | wintergreen prostrate shrub/winters poorly | v | - | v | - | - | - | 2/6 |

### 3.2. Kraków's Residents' Associations with the Smell of Herbs

The results of the survey performed as a part of this study showed that the vast majority of aromatic herbs and their smells were positively perceived by the respondents—residents of Kraków (Figure 2). Although, the smell of catnip *Nepeta* sp. and lovage *Levisticum* sp. were indicated as liked by a smaller group. A number of respondents identified the smell of lovage *Levisticum* sp. as one they did not like, with a similar sentiment expressed for lavender *Lavandula* sp. and salvia *Salvia* sp. An altogether different result in comparison to the previous herbs was returned for the smell of hyssop *Hyssopus* sp., with the majority of respondents reporting this smell to be either completely unknown to them or that it was neither pleasant nor unpleasant. This can mean that although it is a generally well-known plant, as a plant mentioned in the Bible, it was not known to the respondents from direct experience, due to its infrequent use in private or public gardens in Poland.

Almost all of the respondents reported having a favorite aromatic plant, whose smell they saw as particularly pleasant and bringing to mind positive associations. The most important plants to which the respondents ascribed such significance were herbs such as lavender *Lavandula* sp. and mint *Mentha* sp. (Figure 3). There was a slight difference according to the respondents' age. Among young persons between 18 and 29 years of age, the highest number of respondents pointed to lavender *Lavandula* sp., followed by mint *Mentha* sp., which was in turn followed by lemon balm *Melissa* sp., oregano *Origanum* sp., and rosemary *Rosmarinus* sp., which were picked much less often. Among older persons, aged 30 and above, the greatest number of respondents listed their most liked and positively associated herbs as mint *Mentha* sp., followed by lavender *Lavandula* sp., rosemary *Rosmarinus* sp., and lovage *Levisticum* sp., respectively.

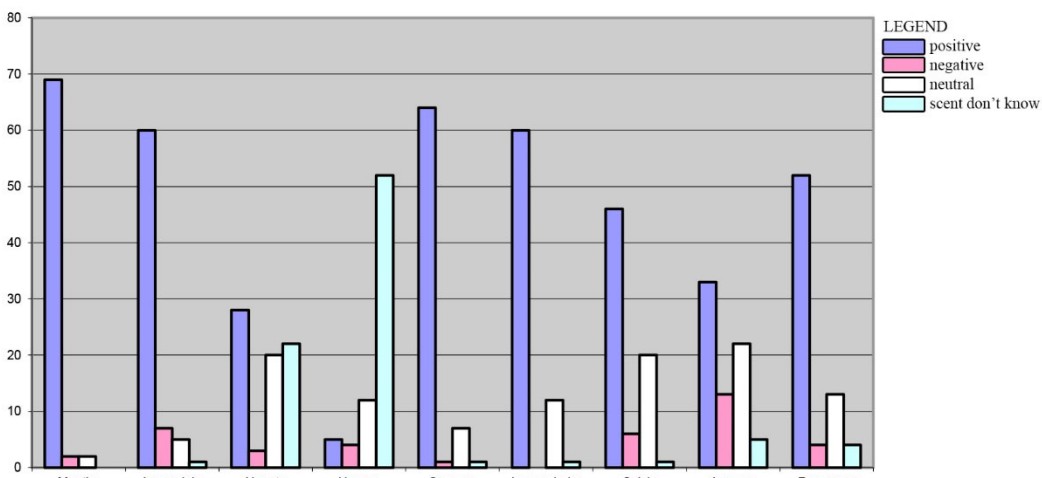

**Figure 2.** Results showing the perception of the smell of each herb as reported by respondents.

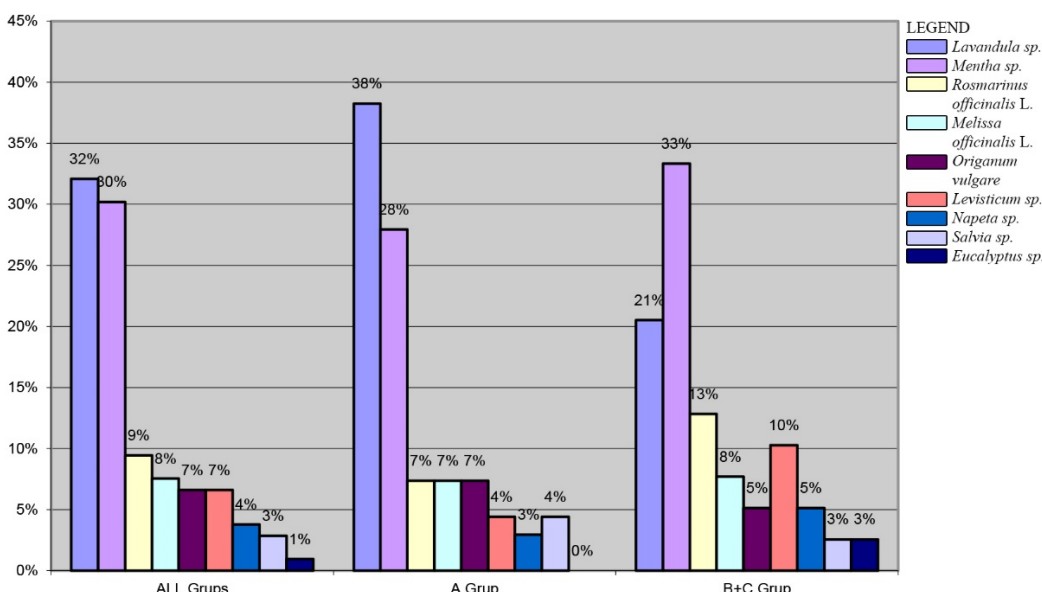

**Figure 3.** Results presenting associations with a smell that held particular significance to respondents divided into two age groups—young people up to 29 years of age (group A) and aged 30 and above (groups B and C).

The respondents reported that the smells in question brought clearly positive associations, primarily linked with home and family, often simultaneously with childhood and youth, the vacation season, as well as memories of exceptional travels, mostly during summer (Table 3). In every age group, positively perceived smells carried at least one of the above meanings. However, a difference in the percentage shares in their ascription to each meaning was observed. In the group with the youngest respondents (group A), the greatest significance was found for associating the smell of one's favorite herbs with home and family (70%), the middle-aged group saw smells associated with home, travel, and summer as equally important, while seniors reported positive associations mostly with home and family (60%).

**Table 3.** Specific associations with past events listed by respondents as tied with preferred aromatic herbs.

|  | ALL GROUPS | AGE GROUP A | AGE GROUP B | AGE GROUP C |
|---|---|---|---|---|
| Home, family (%) | 60 | 70 | 37 | 60 |
| Exceptional travels—often during vacation season, the vacation season (%) | 26 | 21 | 43 | 10 |
| Other—typically described as a personal preference or non-descript (%) | 14 | 10 | 20 | 30 |

In terms of the specific significances that respondents ascribed to the smell of individual herbs (Table 4), most respondents across all age groups reported that it was the smell that was the most important, yet a large group of respondents tied it with the addition of a given herb to a specific beverage or meal. The significance of the application of a given herb as a home remedy and the association of its smell with a preparation intended to aid in minor health disorders was reported to be smaller, yet still substantial.

**Table 4.** The significance of favorite herbs and their smell.

|  | ALL GROUPS | AGE GROUP A | AGE GROUP B | AGE GROUP C |
|---|---|---|---|---|
| The smell of the plant itself (%) | 56 | 61 | 50 | 40 |
| The smell of herbs added to beverages or meals (%) | 30 | 27 | 32 | 40 |
| The smell of a plant as a homemade medicine (%) | 9 | 9 | 7 | 20 |
| The smell of a plant used as an anti-insect agent (%) | 3 | 1 | 7 | 0 |
| Symbolic plant with spiritual significance (%) | 2 | 1 | 4 | 0 |

### 3.3. Therapeutic Potential of Gardens Featuring Aromatic Herbs

The results of the analysis of the therapeutic potential of the gardens under study are presented in Tables 5 and 6 and are graphically illustrated in Figure 4. Most gardens with aromatic herbs were rated to have a potential of around 50%, with the range being 46–59%. Only one garden, near the J. Czapski pavilion, was found to have a slightly higher therapeutic potential of 66%.

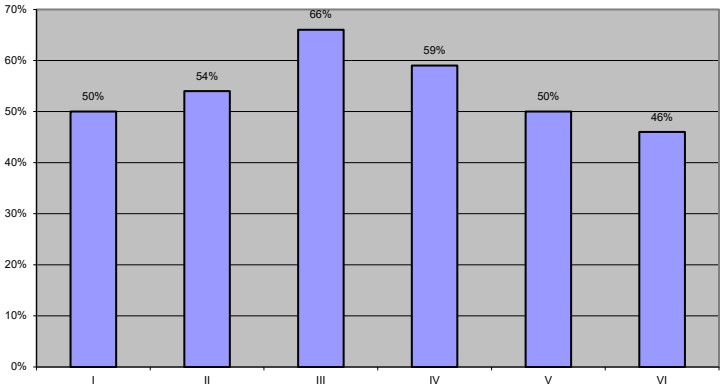

**Figure 4.** Overall results of the therapeutic potential analysis of the gardens under study.

**Table 5.** Analysis of the therapeutic properties of herbal gardens with sensory features, part 1—functional program.

| ATTRIBUTES | | HERBAL GARDENS | | | | FRAGRANT GARDENS | |
|---|---|---|---|---|---|---|---|
| | | 1 | 2 | 3 | 4 | 5 | 6 |
| FUNCTIONAL PROGRAM | | | | | | | |
| A. Enabling physical and psychological re-generation | A-1. Places for rest that facilitate experiencing the surroundings from up close | YES (1) Multiple seats | YES/NO (0.5) None in the garden; behind the pergola there is a meadow with sunbeds for patients | YES/NO (0.5) Seating only on the platform, no other seating | YES/NO (0.5) Near the path adjacent to the garden | YES/NO (0.5) A few benches, some deep inside the garden | YES (1) Numerous seats |
| | A-2. Isolation from the urban environment, noise, smells, and the pressure of time and fast living | YES/NO (0.5) Partially, there is a street just behind a fence | YES (1) The entire garden is walled, the herbal garden is in its corner | YES (1) The garden is in an internal courtyard | YES/NO (0.5) The museum grounds are fenced, the garden is near a fence that separates it from a street | YES (1) The fragrance garden is at the edge of the science garden, located at the side of an extensive park | YES/NO (0.5) Fenced from one side, close to a street from another |
| | A-3. Ability to easily observe animals or people | YES/NO (0.5) Observable insects and birds, herbal plants are up close, but there are taller perennials, bushes, and trees | YES/NO (0.5) Observable insects—the herbs are short, but in large clusters, there are also creepers | YES (1) Observable insects—the plants are in rather tall pots | YES/NO (0.5) Observable insects—the herbs are placed low, and grow in extensive clusters | YES/NO (0.5) Observable insects—many flowering plants are close to the soil, there are also bushes | YES/NO (0.5) The herbs are placed low and often grow in extensive surfaces, there are also creepers and bushes |
| B. Facilitating social contact | B-1. Ability to meet as a group | YES/NO (0.5) Only on the lawn | YES/NO (0.5) On the lawn near the garden, and for patients—on the lawn with sunbeds | YES (1) There is a dedicated meeting space— there are chairs and tables on the terrace, which can be moved | YES/NO (0.5) Only on the lawn near the outer edge of the garden | YES/NO (0.5) There is a small space with benches, but the main footpath crosses it | YES (1) There is a separate square-shaped interior with benches all around; There is another space under a pergola |

**Table 5.** *Cont.*

| | ATTRIBUTES | HERBAL GARDENS | | | | FRAGRANT GARDENS | |
|---|---|---|---|---|---|---|---|
| | | 1 | 2 | 3 | 4 | 5 | 6 |
| C. Facilitating physical activity | C-1. Places for play and recreation | NO (0) The garden itself offers no such capacity; it serves other purposes | NO (0) The herbal garden is not intended for this, but there are other gardens nearby | YES/NO (0.5) There is a lawn, but it is not dedicated to physical exercise | NO (1) There is a lawn, but it is not intended for physical exercise due to the presence of the museum | YES (1) Numerous grassy spaces with devices to perform experiments nearby | NO (0) The garden is not intended for this purpose |
| | C-2. Place dedicated to gardening classes or hortitherapy | NO (0) Absent | NO (0) Absent | NO (0) Absent | NO (0) Absent | NO (0) Absent | NO (0) Absent |
| D. Meeting essential user needs | D-1. Safety in the garden space | YES (1) The area is adjacent to the museum and is fenced | YES (1) The garden is in a fenced area | YES (1) The garden is adjacent to the museum, in a courtyard | YES (1) The garden is adjacent to the museum, it is surrounded by a tall fence | YES (1) The garden is located within a fenced science garden | YES/NO (0.5) There is a street nearby shielded by a row of trees |
| | D-2. Safety in direct contact with plants | YES/NO (0.5) Many of the plants are safe, but some are not | YES/NO (0.5) Numerous safe plants: there are rose bushes | YES (1) Safe plans were used, primarily herbs | YES (1) Most plants are safe | YES/NO (0.5) Many of the plants are safe, there are rose bushes | YES (1) Most plants are safe |
| | D-3. Seating or shelter | YES (1) Shelter inside the museum | YES (1) Shelter under a pergola and in a gazebo near the garden | YES (1) Shelter inside a coffee shop with a terrace inside the garden | YES (1) Shelter in the museum | YES/NO (0.5) Shelter under tree canopies nearby | YES/NO (0.5) Shelter under a pergola, no other shelters |
| | D-4. Sunny and shaded places | YES (1) Both types of places are present | YES (1) There are sunny places and a shaded space under a pergola | YES (1) Both types of places are present | YES/NO (0.5) Both types of places are partially present | YES/NO (0.5) The benches are not shaded in any way | YES (1) Places in the sun and in the shade—under a pergola and tree canopies |

Table 5. *Cont.*

| ATTRIBUTES | | HERBAL GARDENS | | | | FRAGRANT GARDENS | |
|---|---|---|---|---|---|---|---|
| | | 1 | 2 | 3 | 4 | 5 | 6 |
| | D-5. Amenities for the disabled | NO (0) No dedicated amenities for the disabled | YES/NO (0.5) Wide, smooth paths; no other amenities | YES/NO (0.5) Wide paths; no dedicated amenities | NO (0) Absent | NO (0) Aggregate footpath with narrow paved strips, not intended for wheelchair use | YES/NO (0.5) The paths are wide and smooth; no amenities for the visually impaired |
| | D-6. Elements that indirectly affect comfort of use: access to food and drink, toilets, and others | YES (1) Inside the museum | NO (0) None in the immediate vicinity | YES (1) There is a coffee show with a garden view | YES (1) Inside the museum | YES (1) Elements are present nearby, in the park | NO (0) Absent |
| E. Cognitive support | E-1. Features that facilitate education in the garden | YES/NO (0.5) Only plaques with plant names | YES (1) There are plaques with educational content and plant names | NO (0) | YES (1) There are plaques with educational content and plant names | YES (1) There are plaques with plant and genus names, as well as information on origin | YES (1) There are plaques with plant names and information about them; there is a garden plan |
| Score | | 7.5/13 | 7.5/13 | 9/13 | 8.5/13 | 8/13 | 7.5/13 |

Table 6. Analysis of the therapeutic properties of herbal gardens with sensory features, part 2—other attributes.

| ATTRIBUTES | | HERBAL GARDENS | | | | FRAGRANT GARDENS | |
|---|---|---|---|---|---|---|---|
| | | 1 | 2 | 3 | 4 | 5 | 6 |
| FUNCTIONAL STRUCTURE | | | | | | | |
| A. Functional–spatial structure | A-1. Isolation of the garden from its surroundings, creating a separate, intimate space | YES/NO (0.5) From one side, the fence is see-through, with a street alongside it. The fence is partially covered in vines | YES (1) The garden is surrounded by a tall wall | YES (1) The garden is in an internal courtyard surrounded by walls | YES (1) The garden is close to a fence entirely covered in vines; the street nearby is obscured | YES/NO (0.5) The garden is near the outer edge of a science garden, there are no tall insulation plants | YES/NO (0.5) There is a parking lot nearby, a street runs alongside it and is visible despite the presence of a row of trees |

**Table 6.** *Cont.*

| | ATTRIBUTES | HERBAL GARDENS | | | | FRAGRANT GARDENS | |
|---|---|---|---|---|---|---|---|
| | | **1** | **2** | **3** | **4** | **5** | **6** |
| | A-2. Siting in a place that retains fragrances and sounds inside the garden | YES/NO (0.5) Only in some places | YES (1) A tall wall shields from the wind and produces a quiet place | YES (1) The plants in pots receive adequate sunlight, the garden is surrounded by a solid fence and buildings | YES/NO (0.5) Fragrances are retained fully, but sounds only partially | YES (1) Behind the garden there is a small, elevated plateau that shields it from the north, there are also places with shrubs | YES/NO (0.5) Partially |
| Score | | 1/2 | 2/2 | 2/2 | 1.5/2 | 1.5/2 | 1/2 |
| INTERNAL SPACE AND ARCHITECTURAL FORM DESIGN | | | | | | | |
| A. Internal space and architectural form design | A-1. Garden complexity, presence of various garden interiors, proper path system | YES/NO (0.5) Diverse interiors, the paths form loops; there are places for rest, there are no isolated interiors | YES/NO (0.5) Simple layout: there is a pergola with vines | YES/NO (0.5) Simple layout: a path runs around a lawn; plant pots only from one side | YES/NO (0.5) Intriguing garden layout that references a historical model, but the garden is small and has no distinct interiors | YES/NO (0.5) The path has an interesting course, but there are no distinct interiors that would allow for longer stays | YES (1) Interesting sensory garden layout, there are pergolas with resting places, there is a separate interior outside of the main area |
| | A-2. Legibility of composition | YES (1) The composition is legible | YES (1) The composition is legible | YES (1) The composition is geometric and legible | YES (1) The composition is geometric and legible | YES/NO (0.5) The composition is not fully legible, the path has dead ends | YES (1) The composition is geometric and legible |
| | A-3. Presence of water, especially water in motion | NO (0) None | YES/NO (0.5) There are none in the garden, but there is a stream nearby | NO (0) None | NO (0) None | NO (0) None | NO (0) None |

**Table 6.** *Cont.*

| ATTRIBUTES | | HERBAL GARDENS | | | | FRAGRANT GARDENS | |
|---|---|---|---|---|---|---|---|
| | | **1** | **2** | **3** | **4** | **5** | **6** |
| A-4. Plant sensory impact on each of the senses | | YES (1) Numerous senses are stimulated, but the plants cannot be tasted | YES (1) All five senses are stimulated | YES (1) The herbs provide diverse stimuli and can be tasted | YES (1) Numerous senses are stimulated, but the plants cannot be tasted | YES/NO (0.5) It is primarily a fragrance garden | YES (1) Numerous senses are stimulated, but the plants cannot be tasted |
| A-5. Intensity of plant sensory impact (e.g., diversity of species, large spaces, elevated beds) | | YES (1) The species are diverse | YES (1) Large herbal plant beds, there is a pergola with fragrant vines | YES (1) The plants are elevated and placed in large pots | YES (1) the herbs are placed in large groups, the path surface is lined with fragrant thyme | YES (1) Large surfaces covered with aromatic, closely placed plants, along almost the entire path | YES (1) A diverse range of sensory stimuli: there are fragrant herbs, creepers, and bushes |
| A-6. Other sensory active elements (e.g., labyrinth, sensory path) | | NO (0) None | NO (0) None | NO (0) None | YES/NO (0.5) There is a stone path inlaid with joints inlaid with thyme | YES (1) There is a sensory path and labyrinths close to the fragrant garden | NO (0) None |
| Score | | 3.5/6 | 4/6 | 3.5/6 | 4/6 | 3/6 | 4/6 |
| PLACEMAKING | | | | | | | |
| A. Placemaking— personalization and animation of the space, art in the garden and special uses | A-1. Ability to personalize the space | NO (0) None | NO (0) None | NO (0) None | NO (0) None | NO (0) None | NO (0) None |
| | A-2. Ability to animate the space | YES/NO (0.5) A lecture on the history of medicine and herbs can be organized here | NO (0) | YES (1) Multimedia presentations, films on the museum's wall (seats on the grass) | YES/NO (0.5) A lecture on the plants from the work by Marcin of Urzędów can be organized | YES/NO (0.5) Open-air lessons for children, the youth and adults can be organized | NO (0) |

**Table 6.** *Cont.*

| | ATTRIBUTES | HERBAL GARDENS | | | | FRAGRANT GARDENS | |
|---|---|---|---|---|---|---|---|
| | | **1** | **2** | **3** | **4** | **5** | **6** |
| | A-3. Artistic creations | NO (0) None | NO (0) None | YES (1) The museum wall features an artistic exhibition | YES/NO (0.5) There is an ornamental wall that references the garden's style | NO (0) None | NO (0) None |
| | A-4. Special indications for use | NO (0) None | NO (0) None | YES (1) The herbs can be tasted | NO (0) None | NO (0) None | NO (0) None |
| Score | | 0.5/4 | 0/4 | 3/4 | 1/4 | 0.5/4 | 0/4 |
| SUSTAINABILITY | | | | | | | |
| A. Sustainability criteria | A-1. Biodiversity preservation: use of domestic plant species and plants attractive to various groups of animals, creating habitats for animals | YES (1) Some of the plants are attractive to insects; domestic plants; densely planted specimens of varying height offer room for habitats | YES/NO (0.5) Certain plants may be attractive to butterflies or other insects | YES/NO (0.5) Deliberate application of species attractive to hymenoptera | YES (1) Certain plants are attractive to insects; some of the plants are domestic species common in the Sandomierz area | YES/NO (0.5) Some plants are attractive to insects; domestic plants | YES/NO (0.5) Certain plants can be attractive to butterflies and other insects |
| | A-2. Sustainable water management, e.g., stormwater collection and use | NO (0) None | NO (0) None | NO (0) None | NO (0) None | NO (0) None | NO (0) None |
| | A-3. Natural energy sources | NOT APPLICABLE No electrical appliances | NOT APPLICABLE No electrical appliances | NO (0) None | NOT APPLICABLE No electrical appliances | NOT APPLICABLE No electrical appliances | NOT APPLICABLE No electrical appliances |

**Table 6.** *Cont.*

| ATTRIBUTES | | HERBAL GARDENS | | | | FRAGRANT GARDENS | |
|---|---|---|---|---|---|---|---|
| | | 1 | 2 | 3 | 4 | 5 | 6 |
| | A-4. Natural garden mainte-nance methods | No data available | YES (1) Yes, the plants can be harvested | YES (1) | No data available | No data available | No data available |
| Score | | 1/2 | 1.5/3 | 1.5/4 | 1/2 | 0.5/2 | 0.5/2 |
| Total score | | 13.5/27 | 15/28 | 19/29 | 16/27 | 13.5/27 | 12.5/27 |

### 3.4. Water in Sensory Gardens Featuring Herbs

Due to global warming and the existence of the urban heat island effect, along with the occurrence of various associated problems concerning urban greenery and their impact on human health and wellbeing, we investigated the role of water in city gardens. This is why the survey performed as part of this study included questions on whether water and its sound can have a positive impact on the perception of gardens that include herbs and the significance it could have. Most respondents reported that it can have a positive impact to humans and city-dwelling animals—see Table 7.

**Table 7.** Problems associated with water in sensory gardens—survey responses.

|  | **Respondent Responses** |
|---|---|
| -Is the presence of water in a garden, e.g., as a fountain, important? | Yes—62<br>No—5<br>I have no opinion—6 |
| -What does the presence of water mean to you? | Relaxation—26<br>Humidification—5<br>Relaxation and humidification—25<br>Water source for birds and insects—2<br>Others—3<br>It does not mean anything—8 |
| -Where should the water used in a garden come from? | Harvested stormwater and rainwater—62<br>From the grid—2<br>It is irrelevant—9 |

In the light of the problems caused by water deficits and those of cities, the respondents were asked whether they found the source of water used in a garden to be significant. The majority of the respondents replied that they found it significant and that they believed that rainwater and stormwater should be the main source of water used in gardens.

## 4. Discussion

In gardens with sensory features, especially those that feature aromatic herbs, the impact of climate change, and especially global warming, has already become somewhat noticeable. Plants that were rarely seen in urban spaces in the twentieth century due to their possible freezing in the Polish climate [42] are currently more often encountered in cities. One such case is true lavender *Lavandula angustifolia* Mill., which in Poland was first introduced into private gardens, and in recent years it has increasingly been planted in public gardens, near parking lots and on church grounds. It was observed that this species was present in all of the herbal gardens investigated in this study. Rosemary *Rosmarinus officinalis* L. is another such example. It was seen as a pot plant in the Polish climate, originally present on the edges of the Mediterranean Sea, and is typically cultivated in European countries with mild winters [43]. This species has only recently been introduced in Polish public spaces, on a much smaller scale than lavender, as it is a more sensitive species. Rosemary was introduced into the garden near the J. Czapski Museum, which has sensory garden features. In Kraków's city center, the urban heat island effect has been observed, while along the area's edges, temperatures are not always higher than in the surrounding rural areas. This depends on a range of factors [23]. The aforementioned area near the Museum is located in the heart of the city, where temperatures are the highest. In terms of aromatic plants, high temperatures make the fragrances produced by them more intense. The area is also shielded from cross-ventilation by buildings, while from the north it is fenced by an insolated garden wall that is warmed by the sun. These are factors that contribute to the presence of favorable conditions for sensitive plants, including in winter. Rosemary is able to survive winter in such a setting without cold protection, which was observed for several consecutive years. As an evergreen plant, it greatly enlivens garden spaces with its vivid green color. Shanahan et al. argued that the visual features of a green

environment and the biophysical parameters of greenery, stemming from the features of plant structure and physiological activity, both play a part in affecting human health [25].

Expanding the set of plants used in sensory gardens to include species that were previously seen as vulnerable, and which are familiar to and liked by city residents can enhance health benefits. A study by Bengtsson and Grahn [9] demonstrated that a significant role in rehabilitation and convalescence is played by high species diversity in therapeutic gardens, especially during the later stages of the process. These observations can also be applied to city parks, which is why apart from plants introduced by people, it is important to note factors that can contribute to the diversity of domestic species in park cover [44,45]. Many domestic plants or aromatic herbs are attractive to insects and observing animals in therapeutic gardens enhances the scope of sensory stimuli [46] and supports therapy [47]. A study by Cooper Marcus et al. [48] also showed just how important the inclusion of animals is in improving wellbeing. It presented how observing wildlife, including insects and birds, was identified as a major activity among hospital gardens.

The findings indicate that most city residents can be said to positively perceive the smells of various herbs. In addition, regardless of age, many residents expressed that they preferred herbal smells and reported that such smells elicited personal and highly positive associations. In our study, the respondents' favorite aromas were typically associated with the home, childhood, as well as the vacation season, representing a time of rest and relaxation. Winterbottom and Wagenfeld argued that bringing to mind pleasant past experiences can be significant to people who have had to leave their place of residence for various reasons such as illness, homelessness, as well as for people who find themselves in a completely alien environment [3]. The period of childhood, linked with nature or specific plants, remains strongly embedded in a person's memory, as observed by Lohr et al., who observed that childhood experiences can impact one's attitude towards wildlife and gardening in adulthood [49]. Other researchers also pointed to the significance of memories tied with garden settings, as memories of previously encountered gardens were observed in the elderly [17]. The results of our survey clearly showed that aromas can also bring to mind summertime relaxation and pleasant memories of vacation trips. The application of plants such as lavender or rosemary in Central European climate conditions, despite previously not being recommended, can stimulate these distinct memories, especially of travel to warmer areas of Europe.

Smells that bring to mind positive associations, when accounting for clearly pleasant memories, can improve the mood and sense of wellbeing of those who experience them. In our survey, a significant percentage of respondents, who were big-city residents, associated smells with good memories, which means that it can be argued that sensory gardens that stimulate the sense of smell of residents of big cities can improve the quality of life for individuals in a city and their overall wellbeing. Winterbottom and Wagenfeld [3] observed that the smell of lavender *Lavandula* sp., rose *Rosa* sp. and salvia *Salvia* sp. improved mood and eased stress, while the smell of mint *Mentha* sp. and citruses were perceived as invigorating or overwhelming. Lavender and mint were often identified in our study as a respondent's preferred plant, which brought to mind highly positive associations. However, salvia, despite reactions to its smell being reported as either positive or indifferent, was identified as personally significant or bringing to mind special memories by very few respondents. These results are important indicators for designers of therapeutic sensory gardens intended for construction in urban settings.

Based on the survey's results, we also noted that a significant portion of the respondents tied their favorite smells with beverages and food overall. This may indicate that adding such herbs to meals served in cafes operating alongside sensory gardens can enhance and extend the positive effect that appears as a result of a good memory and can thus result in greater therapeutic benefits and an elevated sense of wellbeing. Trojanowska argued that access to food and drink should be an attribute of a park with therapeutic features [39]. There are cases of Polish sensory gardens that feature a coffee shop, such as the garden near the J. Czapski Museum in Kraków.

Our analysis of Polish herbal gardens with sensory features using the Trojanowska method [39,40] as adapted by Krzeptowska-Moszkowicz et al. [6] demonstrated that such gardens have a medium therapeutic potential. It also revealed their limitations. One major problem in cities may be the existence of poor-quality greenspaces, which is a factor that contributes to human health, as it was observed that in urbanized areas such greenery, despite being present, may not impart as many health benefits as good-quality greenspace [50]. The higher adaptability of such gardens to disabled persons (attribute 1: D-5) is also notable, as sensory gardens can be visited by this demographic. The potential personalization of space (attribute 4: A-1) was also found to be overlooked and may prevent visitors from forming a deeper connection with a garden.

In the herbal gardens under analysis, the absence of water features was a major problem (attributes 3: A-3 and 5: A-2). Panel paintings depicted medieval European gardens as equipped with fountains, including in settings with aromatic plants [51]. Water is a significant element in therapeutic gardens and its application, particularly when it is in motion, greatly enhances their therapeutic potential [3,48]. Water is an excellent tool of expanding sensory stimuli, especially auditory, visual, and tactile ones [3]. Most respondents were of the opinion that in a garden with aromatic plants, sounds associated with water, e.g., produced by a small fountain, would be perceived positively. They also noted the positive impact of water on a garden's microclimate, e.g., by humidifying the air and lowering temperature. In our opinion, this highlights the existence of a need for flowing water among city residents and shows that water can positively affect people during periods of high temperature that occur in cities during summer.

The growing water deficit linked to climate change and progressively increasing global warming and water evaporation leads to the problem of supplying plants with water in agriculture and horticulture, and it is not limited to Poland [21]. It also concerns urban sensory gardens, wherein the urban heat island effect further escalates the situation [23,52]. Our survey revealed that the majority of the surveyed residents of a big city, in this case Kraków, found the source of water used to irrigate gardens to be a significant issue. Most respondents expressed the opinion that harvesting stormwater and rainwater for this purpose would be preferable to the use of grid-sourced water. This may indicate that city residents have an awareness of contemporary water-accessibility problems, the significance of water overall and pro-environmental solutions in this regard.

**Author Contributions:** Methodology, I.K.-M.; writing—original draft preparation, I.K.-M.; writing—review and editing, Ł.M. and K.P.; visualization, Ł.M. and K.P. All authors have read and agreed to the published version of the manuscript.

**Funding:** This research received no external funding.

**Data Availability Statement:** Not applicable.

**Acknowledgments:** We would like to thank the reviewers and editor for their accurate comments and assistance in improving the manuscript.

**Conflicts of Interest:** The authors declare no conflict of interest.

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
