# Peer review of "Urban Sensory Gardens with Aromatic Herbs in the Light of Climate Change: Therapeutic Potential and Memory-Dependent Smell Impact on Human Wellbeing"

_land, doi:10.3390/land11050760_

Round 1
Reviewer 1 Report
This study is very interesting, but with a population of nearly 780,000 people in Kraków, the sample size of the survey is too small. The sample size should be calculated based on the population and further data should be obtained by the authors. Furthermore, there is no statistical analysis in the article.
Author Response
Dear Reviewer,
Thank you very much for your valuable opinion.
In response, we would like to emphasise that the purpose of the survey was to establish general trends of phenomena, preferences and not to establish precise results as in typical survey research. Therefore smaller or bigger statistical error connected with the size of the group of respondents does not play a significant role here. Taking into account the aim of the research and the size of the sample we considered statistical analyses to be not very reasonable.
The article will be a contribution to further, more detailed research using the methods used and then most likely extended statistical data will appear.
Kind regards
Karolina Porada
Reviewer 2 Report
The manuscript is quite well written but it would benefit from being revised by a native English speaker. There are a number of errors that need to be corrected - for example:
line 65 - delete 3. Results
lines 66-68 - delete
line 69 - the urban tissue
line 101 - are a (not area)
Table 1 and 2 - FEATURING (not FEATURIN)
line 185 - three concewrning (not three-concerning)
Page 8 - Figure 1 should be Figure 2 as you already had Figure 1 previously (the map). The colours in the key are not clear
Page 8 - Figure 2 should be Figure 3. The key etc. needs to be translated
Page 15 - Figure 3 - is there meant to be a figure here? If so add the figure and change the caption to Figure 4; alternatively delete
line 347 - Cooper Marcus (not Cooper, Marcus)
Author Response
Dear reviewer,
Thank you very much for pointing out to us the errors in the text. They have all been corrected according to your indications:
line 65 - delete 3. Results - deleted
lines 66-68 - delete - deleted
line 69 - the urban tissue - corrected
line 101 - are a (not area) - typing error, corrected
Table 1 and 2 - FEATURING (not FEATURIN) - typing error, corrected
line 185 - three concewrning (not three-concerning) - typing error, corrected
Page 8 - Figure 1 should be Figure 2 as you already had Figure 1 previously (the map). The colours in the key are not clear - numbers of figures corrected, we also added new legends
Page 8 - Figure 2 should be Figure 3. The key etc. needs to be translated - numbers of figures corrected, we also added new legends
Page 15 - Figure 3 - is there meant to be a figure here? If so add the figure and change the caption to Figure 4; alternatively delete - the figure was added
line 347 - Cooper Marcus (not Cooper, Marcus) - typing error, corrected
Under the influence of the other two reviews, the article has also changed the array.
Yours sincerely
Karolina Porada
Reviewer 3 Report
The paper addresses an interesting topic related to the importance of herbs gardens. I found the paper incoherent between aim, proposed methodology and results. I found no relevance of proposed analyses for climate changes. Why would the authors to complicate the paper, having no methodology and no results considering the climate change? I found that the main message of the paper has no relationship with climate change, but with the people perception related to herbs gardens. In my view there are enough results and enough contribution in the field. So, my suggestion is to be more direct in paper design, addressing directly the topic of herbs gardens connected with people perception and their contribution for ecosystem services provision.
Author Response
Dear Reviewer,
The authors agree with the reviewer's remark that the primary theme of our work is the analysis of sensory gardens in terms of their therapeutic importance and the effects of different aromatic plants on the different recipients of these gardens, and that the impact of climate change is a secondary concern in this case. In our opinion, however, it is an important issue when it comes to urban gardens, which cannot be completely ignored because it affects the character of contemporary gardens.
Following the above reviewer's advice, changes have been made to the text of the work, especially in terms of marking the correct purpose, in line with what the work is about. So that it corresponds to the actual core of the work and the most strongly addressed issues.
We hope for another positive evaluation of our work.
Yours sincerely
Karolina Porada
Round 2
Reviewer 1 Report
The goal of this research was to look at the plants that were used in urban sensory gardens that contained aromatic herbs. The impact of climate change, especially higher temperatures, on the character of contemporary urban gardens was considered in the study. The paper is still in its early stages and needs more work before it can be published.
The data is incomplete because the sample size is too small. It is strongly advised that authors consult with a statistician. The discussion section requires extensive rewriting; please compare it to those from other studies. You should also include study limitations and future research.
Author Response
Dear Reviewer,
Thank you for reviewing our work again and taking the time to do so. Thank you very much for your comment and we agree with it. Due to the limited time we currently received from the publisher to revise the paper, we are not able to increase the sample group or rewrite the discussion section. However, we have decided to highlight the limitations of the study. We have added the following text in lines 179-181:
The sample group is small and therefore cannot be considered representative, but in our opinion is sufficient for the preliminary study described. In future research the target group of respondents will be expanded.
Yours sincerely
Authors
Reviewer 3 Report
Congratulation for your work.
Author Response
Dear Reviewer,
Thank you for reviewing our work again and taking the time to do so.
Yours sincerely
Authors